# Suppression of Root Rot Fungal Diseases in Common Beans (*Phaseolus vulgaris* L.) through the Application of Biologically Synthesized Silver Nanoparticles

**DOI:** 10.3390/nano14080710

**Published:** 2024-04-18

**Authors:** Ezzeldin Ibrahim, Abdelmonim Ali Ahmad, El-Sayed Abdo, Mohamed Ahmed Bakr, Mohamed Ali Khalil, Yasmine Abdallah, Solabomi Olaitan Ogunyemi, Mohamed Mohany, Salim S. Al-Rejaie, Linfei Shou, Bin Li, Anwar A. Galal

**Affiliations:** 1State Key Laboratory of Rice Biology and Breeding, Ministry of Agriculture Key Laboratory of Molecular Biology of Crop Pathogens and Insects, Key Laboratory of Biology of Crop Pathogens and Insects of Zhejiang Province, Institute of Biotechnology, Zhejiang University, Hangzhou 310058, China; ezzelbehery8818@yahoo.com (E.I.); 0622251@zju.edu.cn (S.O.O.); 2Department of Vegetable Diseases Research, Plant Pathology Research Institute, Agriculture Research Centre, Giza 12916, Egypt; mohamedalikhalel@gmail.com; 3Department of Plant Pathology, Faculty of Agriculture, Minia University, El-Minia 11432, Egypt; abdelmonim.ali@mu.edu.eg (A.A.A.); elsayedabdou@gmail.com (E.-S.A.); yasmeen.abdallah@mu.edu.eg (Y.A.); anwar_galal@yahoo.com (A.A.G.); 4Department of Self-Pollinated Vegetable Crops, Horticulture Institute, Agriculture Research Centre, Giza 12916, Egypt; mohahmbak@gmail.com; 5Department of Pharmacology and Toxicology, College of Pharmacy, King Saud University, P.O. Box 55760, Riyadh 11451, Saudi Arabia; mmohany@ksu.edu.sa (M.M.); rejaie@ksu.edu.sa (S.S.A.-R.); 6Station for the Plant Protection & Quarantine and Control of Agrochemicals of Zhejiang Province, Hangzhou 310004, China

**Keywords:** root rot fungi, silver nanoparticles, antifungal activity, biosynthesis, plant extracts

## Abstract

The biosynthesis of silver nanoparticles (AgNPs) using plant extracts has become a safe replacement for conventional chemical synthesis methods to fight plant pathogens. In this study, the antifungal activity of biosynthesized AgNPs was evaluated both in vitro and under greenhouse conditions against root rot fungi of common beans (*Phaseolus vulgaris* L.), including *Macrophomina phaseolina*, *Pythium graminicola*, *Rhizoctonia solani*, and *Sclerotium rolfsii*. Among the eleven biosynthesized AgNPs, those synthesized using *Alhagi graecorum* plant extract displayed the highest efficacy in suppressing those fungi. The findings showed that using AgNPs made with *A. graecorum* at a concentration of 100 μg/mL greatly slowed down the growth of mycelium for *R. solani, P. graminicola*, *S. rolfsii*, and *M. phaseolina* by 92.60%, 94.44%, 75.93%, and 79.63%, respectively. Additionally, the minimum inhibitory concentration (75 μg/mL) of AgNPs synthesized by *A. graecorum* was very effective against all of these fungi, lowering the pre-emergence damping-off, post-emergence damping-off, and disease percent and severity in vitro and greenhouse conditions. Additionally, the treatment with AgNPs led to increased root length, shoot length, fresh weight, dry weight, and vigor index of bean seedlings compared to the control group. The synthesis of nanoparticles using *A. graecorum* was confirmed using various physicochemical techniques, including UV spectroscopy, Fourier-transform infrared spectroscopy (FTIR), transmission electron microscopy (TEM), X-ray diffraction (XRD), scanning electron microscopy (SEM), and energy-dispersive X-ray spectroscopy (EDS) analysis. Collectively, the findings of this study highlight the potential of AgNPs as an effective and environmentally sustainable approach for controlling root rot fungi in beans.

## 1. Introduction

The world population is expected to reach nine billion by 2050, and more agricultural production will be needed [1,2]. The common bean, also known as *Phaseolus vulgaris* L., is a legume with great nutritional value, and many of the world’s population depend on it as their food source, which makes it ranked third in terms of cultivated area after soybeans and peanuts [3,4]. Moreover, the planted beans gradually increased year after year, from 23 million hectares in 1999 to 36 million hectares in 2018 [5]. This led to an increase in production from 15 million tons to 31 million tons. In addition to its main role in meeting human nutritional needs, *P. vulgaris* is considered to play a major role in improving soil fertility due to its association with some bacterial species of the genus Rhizobium, which can fix nitrogen that benefits subsequent crops and increases their productivity [6]. Unfortunately, it is estimated that about one-third of the world’s crop production is lost each year due to plant pathogens [7] and that single pathogenic fungi cause 20–40% of crop loss [8]. Also, beans are affected by many fungal diseases that attack them, especially the fungi that cause root rot, stem blight, and stem rot, causing severe economic damage [9,10,11]. These diseases, which cause significant losses exceeding 50% of bean crop production [12], are primarily caused by *Macrophomina phaseolina, Pythium* spp., * Rhizoctonia solani*, and *Sclerotium rolfsii* [5,13,14,15]. 

The use of fungicides is currently a major strategy for controlling these fungi. However, the intended outcome of fighting these illnesses was not realized. Furthermore, overuse of fungicides damages ecosystems and humans and results in the creation of new strains of organisms that are resistant to the chemicals [12,16]. Therefore, there is a need for alternative control methods to fight these diseases and limit their spread and damage. Recently, nanoparticles have attracted the attention of many scientists working in a variety of disciplines, including those in the agricultural sector interested in plant disease suppression due to their unique properties [17,18,19]. Many of them have observed that silver nanoparticles (AgNPs) have a distinct role against a wide range of microorganisms, such as viruses, bacteria, and fungi, which infect and damage plants [20,21,22,23,24]. Moreover, AgNPs showed potent antifungal activity against various pathogens, including *Trichoderma* spp., *Candida albicans, C. tropicalis, Fusarium oxysporum, Trichosporona sahii, Aspergillus niger, Rhizoctonia solani, Curvularia lunata, Colletotrichum* spp., *Magnaporthe oryzae*, and *Fusarium* spp. [25,26]. This antimicrobial effect is attributed to multiple mechanisms, such as AgNPs accumulating on the cell walls of pathogens, leading to structural damage, as well as inducing the production of reactive oxygen species (ROS) [27]. Furthermore, AgNPs have shown promise for enhancing plant growth [22]. However, the use of AgNPs raises several concerns, with the foremost being their potential environmental impact. AgNPs can accumulate in soil, posing a threat to nontarget beneficial organisms and aquatic life [28]. This accumulation has the potential to disrupt the delicate environmental balance and disturb microbial ecosystems [29]. Also, using AgNPs without thinking might lead to the development of more dangerous and resistant pathogens, which would make AgNPs less useful in the long run for controlling diseases. Furthermore, the exposure of both humans and animals to AgNPs poses potential health risks. Excessive use of AgNPs, particularly if they enter the food chain, could have adverse effects on human health [30,31].

AgNPs can be synthesized using various methods, including physical, chemical, and biological approaches [27,32,33,34]. However, biological methods are considered safer and more environmentally friendly compared to conventional physical and chemical methods [13,14]. The biological synthesis process is simple, involving a single vessel setup, and it is rapid, cost-effective, and eco-friendly. Biosources, such as plant extracts, contain polyphenols and proteins that act as reducing agents, thereby reducing the need for hazardous external chemical-reducing agents and minimizing toxicity. Moreover, the green synthesis approach eliminates the requirement for additional capping agents, further reducing costs and simplifying the synthetic process. In contrast, chemical and physical methods are often limited in large-scale applications, as they can be costly, energy-intensive, time-consuming, and challenging in terms of waste removal [35]. Utilizing plant extracts, including those derived from fruits, flowers, and seeds, for AgNP synthesis represents a promising biological approach that avoids the production of hazardous compounds [24]. Therefore, incorporating plant extracts as a biological agent for nanoparticle synthesis can address the aforementioned concerns and potentially play an effective role in reducing plant pathogens. Therefore, this study aims to use plant extracts to synthesize AgNPs and evaluate their antifungal activities against diseases of root rot in beans.

## 2. Materials and Methods

### 2.1. Fungal Source and Growth Conditions

The four phytopathogenic fungi employed in this study were *M. phaseolina, P. graminicola, R. solani*, and *S. rolfsii*, collected from the Vegetable Pathology Research Department, Plant Pathology Research Institute, Agricultural Research Center, Giza, Egypt. Fungi were cultured on potato dextrose agar (PDA), which had a pH of 7 and was made up of 16 g of agar, 200 g of dried potato infusion, and 20 g of dextrose per liter. 

### 2.2. Pathogenicity Test

#### 2.2.1. In Vitro Pathogenicity Test 

The pathogenicity ability of *M. phaseolina, P. graminicola, R. solani*, and *S. rolfsii* was tested by radicle assay in Petri dishes, according to [36], with some modifications. In brief, isolates were grown at 25 ± 2 °C for 5 days. A 5 mm-diameter disc from the fungal cultures’ active growth margins was taken out, put in the center of a Petri plate full of 0.2% water agar, and cultured for three days at 25 ± 2 °C. Bean seeds of the susceptible variety (cv. Nebraska), sterilized with 4% hypooxychloride for 15 min and rinsed with sterile distilled water, were sown in a ratio of four seeds at equal distances around the mycelial disc. As a control treatment, seeds were placed around the sterilized PDA disc. All plates were stored at 25 ± 2 °C for five days, and disease development was rated based on the size of the radicle necrosis area using a scale described by [37]. 

#### 2.2.2. Pathogenicity Test under Greenhouse Conditions

##### Inoculum Preparation and Soil Infestation

Sorghum grains were used for propagation of *M. phaseolina*, *P. graminicola*, *R. solani*, and *S. rolfsii*, according to [38]. Sorghum kernels were first soaked in water for 4 h; excess water was drained, then packed into 250 mL conical flasks at 100 g per flask and sterilized at 121 °C for 35 min. After cooling to room temperature, the flasks were inoculated at a rate of 6 mycelia discs (5 mm in diameter) for every flask of isolates (five days old in culture) separately under sterilized conditions in a laminar airflow cabinet. The inoculated flasks were kept at 26 ± 1 °C for ten days or until fungal growth completely covered the sorghum kernels. Flasks were inoculated with PDA discs as a control. For this, 1% (*w*/*w*) of sorghum inoculum was used to inoculate pots with sterilized soil. Pots inoculated with sorghum without fungi were used as controls. Surface-sterilized bean seeds (cv. Nebraska) were planted in these pots. After seven days of seed planting, the germination rate was recorded. 

##### Virulence Assay in Bean Seedling

For seedling infection, five-day-old bean seedlings (cv. Nebraska) were inoculated as described previously [39], with some modifications. Briefly, the sorghum grain inoculum of all pathogens prepared as explained above was added around the roots at a rate of 10 grains per plant. Inoculated plants were observed daily, and three weeks after inoculation, root disease severity (%), foliar disease severity (%), root length (cm), shoot length (cm), weight of fresh plants (g), and weight of dry plants (g) were measured.

### 2.3. Biosynthesis of AgNPs Using Various Plant Extracts

#### 2.3.1. Preparation of Plant Extract

Eleven extracts were prepared from 11 plant parts (Table 1 and Figure 1), according to [40]. In brief, plant parts were carefully cleaned with double-distilled water (ddH_2_O) and then air-dried. Dried plant parts were ground in a blender, and 5 g of each resulting powder was added to 100 mL of ddH_2_O and placed in a water bath at 60–70 °C for 1 h. After that, the extracts were left to cool at room temperature before their capacity to synthesize AgNPs was assessed.

#### 2.3.2. Formation of AgNPs

AgNP synthesis was performed as described by [41], with several modifications. In separate beakers, 30 mL of each plant extract was combined with 70 mL of a 3 mM AgNO_3_ solution obtained from Sinopharm Chemical Reagent Co., Ltd. (Shanghai, China). The mixture was stirred at 600 rpm and maintained at a temperature of 70 °C with a pH of 7 for 1 h. Subsequently, the resulting nanoparticle solution was centrifuged at 12,000× *g* for 15 min. The supernatant was discarded, and the precipitated nanoparticles were collected, rinsed with ddH_2_O, dried, and stored for further use.

### 2.4. AgNPs Efficacy on the Growth of the Root Rot Fungi

To study the effect of the obtained AgNPs on the growth of *M. phaseolina, P. graminicola, R. solani*, and *S. rolfsii*, the method described by [42] was used. In summary, a 5-day-old fungal disc with a diameter of 5 mm, obtained from the aforementioned fungi, was individually inoculated onto PDA Petri plates containing a concentration of 100 μg/mL for each of the eleven AgNPs. After 5 days of incubation at 25 °C, the radial growth of the fungal mycelium was measured. Among these, the most potent biosynthesis was selected for further characterization and antifungal experiments. 

### 2.5. Characterization of the Biosynthesized AgNPs

Several techniques were used to characterize AgNPs that were produced using aqueous *A. graecorum* leaf extract. UV-visible spectrophotometry was used to examine the visible absorption spectroscopy of AgNPs [17]. Using Fourier-transform infrared (FTIR) and an AVATAR 370 FTIR spectrometer (Thermo Nicolet, Madison, WI, USA)with a resolution of 4 cm in the spectrum range of 4000–500 cm, the functional groups of *A. graecorum* leaf extract responsible for converting Ag ions to AgNPs were discovered. Using scanning electron microscopy (SEM), TM-1000, Hitachi, Japan; transmission electron microscopy (TEM), JEM-1230, JEOL, Akishima, Japan; and energy-dispersive spectrum analysis (EDS), the form, size, and presence of Ag ions in the produced AgNP pellets were ascertained. The X-ray diffraction (XRD) on an XPert PRO diffractometer with a detector voltage of 45 kV and 40 mA and CuKo radiation was used to establish the crystalline nature of biosynthesized AgNPs [42].

### 2.6. Minimum Inhibitory Concentrations of AgNPs 

A 5 mm diameter disc of *M. phaseolina, P. graminicola, R. solani*, and *S. rolfsii* was inoculated into Petri dishes (9 cm in diameter) containing a mixture of PDA medium with AgNPs at four concentrations (25, 50, 75, and 100 μg/mL) in order to identify the minimum inhibitory concentration (MIC) of AgNPs. AgNP-free PDA plates served as the control. Following five days of incubation at 25 °C, the diameter of fungal growth was measured, and the MIC of fungal growth inhibition was computed. 

### 2.7. Inhibition of Seed Infection by Root Rot Fungi Using the MIC of AgNPs

According to [36], the MIC inhibitory effect of silver nanoparticles on root rot diseases of beans was evaluated in vitro. In summary, the *P. vulgaris* (cv Nebraska) seeds were sterilized for 15 min using 4% sodium hypochlorite, and then they were washed three times with sterile ddH_2_O. Sterilized bean seeds were submerged in the MIC of AgNPs for two hours. As a control, bean seeds soaked in ddH_2_O were used. The seeds were sown around fungal discs at a rate of 4 seeds per plate. Germination rate (%) and radical length (mm) were recorded 5 days after sowing.

### 2.8. Antifungal Activity of AgNPs MIC under Greenhouse Conditions 

The MIC-inhibitory effect of AgNPs on root rot diseases in beans in the greenhouse was evaluated according to [43]. Pots with sterilized soil were inoculated with sorghum grain inoculums of *M. phaseolina, P. graminicola, R. solani*, and *S. rolfsii* at 1% (*w*/*w*). Seeds of the Nebraska variety sterilized with sodium hypochlorite, as mentioned previously, were planted at a rate of five seeds per pot after being immersed for four hours in either the MIC of AgNPs or water as a control. Seeds grown in a soil pot containing sterilized sorghum kernels that were not inoculated with pathogens were used as a negative control. The experiment was under daily observation, and at twenty-five days after planting, germination rates (Gr), pre-emergence damping off, post-emergence damping off, root disease severity (Rs), foliar disease severity (Fs), root length (Rl), shoot length (Sl), fresh weight (Fw), dry weight (Dw), and vigor index (Vi) were recorded. A total of 15 seeds (three pots) were used for each strain, and the experiment was repeated two times.

### 2.9. Statistical Analysis

Three replicates of each experiment were used in the fully randomized design of this study. The results were displayed as the mean ± SD (standard deviation). SPSS version 16.0 was used for the statistical analysis (SPSS Inc., Chicago, IL, USA). The results were statistically significant at *p* < 0.05 or 0.01.

## 3. Results and Discussion

### 3.1. Pathogenicity Test

All fungal isolates were pathogenic to beans and caused radical necrosis in vitro. In contrast, no symptoms were observed with the control treatment (Figure 2). The results showed that the rates of radical bean necrosis caused by *R. solani, S. rolfsii, M. phaseolina*, and *P. graminicola* were 63.00, 41.67, 22.33, and 10.00%, respectively. The length of the radical was also varied between fungal isolates. The average radical bean lengths were 40.33, 27.67, 45.00, and 34.67 mm, respectively, after infection with *R. solani, S. rolfsii, M. phaseolina*, and *P. graminicola*, while the average radical length in the control was 72.33 mm. Consistent with our results, the pathogenicity of *R. solani* was tested on chickpeas by radicle assay in a Petri dish under in vitro conditions, and similar results have been reported [36].

According to this study’s findings, the infections decreased the rate at which bean seeds germinated in greenhouse conditions. The germination rates of bean seeds were 33.33%, 13.33%, 0.00%, and 0.00%, respectively, when the soil was infested with *M. phaseolina, R. solani, S. rolfsii*, and *P. graminicola*, respectively, while the germination rates in the control were 100% (Figure 3). The results also showed that these fungi have affected seedling growth. Where there was a decrease in root length, shoot length, fresh weight, and dry weight of bean seedlings after infection with pathogens, rotting symptoms appeared on the stem and roots and led to seedlings dropping (Table 2 and Figure 4). Root rot fungi are among the most common pathogenic diseases in bean-producing areas, attacking bean plants and causing pre- and post-emergent seedling damping, stem rot, and root rot [12,39,44,45].

### 3.2. Biosynthesis of AgNPs

According to our findings, after incubating 30 mL of each plant extract with 70 mL of AgNO_3_ and stirring at 600 rpm and 70 °C for one hour, the solution’s color altered from light yellow to a deep brown, suggesting the creation of nanoparticles in the reaction mixture (Figure 5). According to reports, the presence of certain biomolecules, such as proteins, terpenoids, enzymes, polysaccharides, vitamins, amino acids, phenolic compounds, etc., caused the color change from yellow to brown and refers to the conversion of silver particles into nanoparticles [41,42,46,47].

### 3.3. Efficacy of AgNP Treatment on Root Rot Fungi

By monitoring the radial growth of mycelium, the antifungal activities of eleven AgNPs against root rot fungi (*R. solani, P. graminicola, S. rolfsii*, and *M. phaseolina*) were studied in vitro (Figure 5). As shown in Figure 6, AgNPs reduced the radial growth of the mycelium for all of these phytopathogenic fungi at a concentration of 100 μg/mL. Inhibition of mycelium growth was recorded compared to the control treatments (Table 3). The biosynthesis of AgNPs using *A. graecorum* was the most effective synthetic nanomaterial for suppressing fungal growth. Using 100 μg/mL of biosynthesized AgNPs using *A. graecorum* resulted in a reduction of the mycelial diameter of *R. solani, P. graminicola, S. rolfsii*, and *M. phaseolina* by 92.60, 94.44, 75.93, and 79.63%, respectively. Therefore, we focused on using only this nanomaterial (AgNPs biosynthesis using *A. graecorum*) for further investigations. According to earlier research, AgNPs can be employed as antifungal agents to stop fungal infections in plants, including *Fusarium oxysporum, Magnaporthe oryzae, Phytophthora cinnamomi, Alternaria solani, Aspergillus niger*, and *Aspergillus flavus* [17,18,42,48,49,50]. 

### 3.4. Characterization of AgNPs Synthesized with A. graecorum

The formation of AgNPs was monitored by UV-Vis analysis by collecting the reaction mixture of plant extract and AgNO_3_ at the point of nanoparticle formation. The resulting UV-Vis spectrum showed absorption in the visible range of 400 to 450 nm, with a sharp and intense peak observed at 432 nm (Figure 7), confirming AgNP formation. Conversely, when examining the UV spectrum of phytomolecules derived solely from *A. graecorum* extract, an absorption peak at 280 nm was observed. Previous studies have reported various wavelengths within the range of 400 to 450 nm for AgNPs, including 409 nm, 412 nm, 422 nm, 418 nm, and 430 nm [21,24]. The surface plasmon resonance (SPR) of metallic nanoparticles is known to be sensitive to several factors, such as shape, size, and interparticle interactions, including cluster formation, with the surrounding medium [51].

The biosynthesis of AgNPs using *A. graecorum* was characterized by TEM and SEM analysis, revealing spherical nanoparticles with nanoscale sizes ranging from 4.02 nm to 21.90 nm, and an average size of 8.20 nm (Figure 8A,B). These findings also suggest the presence of organic molecules acting as stabilizing agents on the surface of the silver nanoparticles. The accumulation of these organic molecules may be attributed to hydrogen bonding and/or electrostatic interactions between the functional groups of *A. graecorum* and the AgNP surface. Previous studies have reported similar spherical structures in the biosynthesis of AgNPs using various plant extracts. For instance, AgNPs were synthesized with *Phyllanthus emblica* fruit extract [41]. The EDS results showed that the carbon, silicon, and silver peaks were 19.31, 5.85, and 74.84%, respectively (Figure 8C), and the silver ion peak was formed at 3 KeV. The results are consistent with the silver nanoparticle literature, where an Ag peak was observed at 3 KeV [22].

The crystal nature and particle size characteristics of AgNP biosynthesis were confirmed by XRD analysis (Figure 9A). The AgNPs biosynthesis using *A. graecorum* plant extract showed strong diffraction peaks at 28.81°, 31.98°, 37.85°, 46.03°, and 77.06°, with crystalline silver planes (101), (111), (200), (220), and (311), similar to the results obtained by [41,46]. The additional peaks observed at approximately 28.8°, 31.98°, and 46.03° in the XRD patterns can be attributed to the bio-organic phase present on the surface of the particles. The broadening of peaks in XRD patterns of solids is generally indicative of smaller particle sizes. This broadening effect reflects the influence of experimental conditions on the nucleation and growth of crystal nuclei [52]. Based on the FTIR analysis, the functional groups of the synthesized AgNPs were identified. Figure 9B presents the FTIR spectra comparing the biogenic AgNPs derived from *A. graecorum* leaf extract after reaction with AgNO_3_ to the leaf extract control without AgNO_3_. The spectra exhibit marginal shifts in peak positions, as depicted in Figure 9B. The spectral analysis provides insights into the functional biological groups responsible for stabilizing the nanoparticles and acting as capping or stabilizing agents. The FTIR measurements of AgNPs synthesized using *A. graecorum* leaf extract revealed distinct absorption peaks at specific wavenumbers. These peaks include 3444 cm^−1^, indicating N-H elongation vibrations; 2920 cm^−1^, 1384 cm^−1^, and 1077 cm^−1^, associated with C-H elongation vibrations, C=N binding Amide II, O-H deformation vibrations, and C-N elongation amine vibrations, respectively. The peak at 1637 cm^−1^ represents the carbonyl group C=O and C=C elongation vibrations. Furthermore, the presence of proteins linked to AgNPs through amine groups is indicated by the shift from 1648 cm^−1^ (in leaf extract) to 1637 cm^−1^ (in AgNPs), corresponding to amide I vibrations. Similarly, the peak observed around 1439 cm^−1^ in AgNPs spectra, assigned to C–H symmetric vibrations, shifted from the original 1384 cm^−1^ in the leaf extract. The C–C stretching vibration peak at 1108 cm^−1^ in the extract shifted to 1077 cm^−1^ in AgNPs. Additionally, the N-H elongation vibrations peak at 3371 cm-1 in the extract and shift to 1444 cm^−1^ in AgNPs (Figure 9B). The presence of such groups in the chlorofluorocarbons (CFCs) from the plant extract of *A. graecorum* confirms the presence of proteins and indicates that these functional groups play a major role in reducing Ag^+^ to Ag^0^ [53,54,55].

### 3.5. Minimum Inhibition Concentration (MIC) of AgNPs 

Metal nanoparticles such as magnesium nanoparticles [56,57], copper nanoparticles [58,59], and zinc nanoparticles [60] are widely used to control fungal pathogens in plants. However, silver nanoparticles have strong antifungal activity and are rarely used to control plant fungal pathogens. In this study, silver nanoparticles were biosynthesized using *A. graecorum* to control *M. phasolina, R. solani, S. rolfsii*, and *P. graminicola.* Concentrations of AgNP ranging from 25 to 100 μg/mL were tested as antifungal agents, as shown in Figure 10. The results showed that AgNPs had antifungal activity at concentrations of 25, 50, 75, and 100 μg/mL against *M. phasolina, R. solani, S. rolfsii*, and *P. graminicola*. A 100 μg/mL concentration had the highest antifungal activity with inhibition rates of 73.33, 77.78, 92.95, and 93.33%, with *M. phasolina, S. rolfsii, R. solani*, and *P. graminicola*, respectively. From these results, the MIC was 75 μg/mL, which was the lowest concentration for inhibition of all fungi, with inhibition rates of 40.24, 41.11, 60.00, and 70.00% for *M. phaseolina, S. rolfsii, R. solani*, and *P. graminicola*, respectively.

### 3.6. In Vitro MIC Inhibited Bean Root Rot Diseases

In vitro, the MIC (75 μg/mL) of AgNPs was evaluated to inhibit root rot diseases of beans by measuring germination rates and radicals’ length. It was found that bean seeds grown in dishes with *M. phasolina, S. rolfsii, R. solani*, and *P. graminicola* had much lower germination rates and radical lengths than bean seeds treated with AgNPs at a concentration of 75 μg/mL and grown in dishes with the same pathogens. Particularly, the germination rates were 91.67, 58.33, 0.00, and 33.33%, and the length of radicals was 40.33, 33.00, 0.00, and 2.67 mm, respectively, when untreated bean seeds were grown in inoculated dishes with *M. phasolina, R. solani, S. rolfsii*, and *P. graminicola*. The germination rates were 100.00, 91.67, 66.67, and 91.67%, and the length of radicals was 92.76, 83.35, 40.67, and 59.00 mm, respectively, when the seeds were treated with silver nanoparticles (75 μg/mL) and grown in inoculated dishes with the same pathogens mentioned above. Moreover, seeds treated with AgNPs had a higher growth rate and radical length (Figure 11). Our results showed that the radical length was 110 mm when seeds were treated with AgNPs and 63.00 mm with untreated seeds. In line with what we found, Cu-chitosan nanoparticles were able to kill *Alternaria solani* and *Fusarium oxysporum* fungi and improve the rate at which tomatoes germinated, the length of their roots and shoots, and the vigor index of their seeds [61]. Despite the known effectiveness of silver nanoparticles (AgNPs) against pathogens, several studies have highlighted the emergence of resistance in certain Gram-negative bacterial species. For example, studies have demonstrated that repeated exposure to silver nanoparticles has led to the development of resistance in bacteria such as *Escherichia coli* 013, *Pseudomonas aeruginosa* CCM 3955, and *Escherichia coli* CCM 3954, highlighting the potential for the development of resistance over time [62].

### 3.7. Under Greenhouse MIC Inhibited Bean Root Rot Diseases

Under greenhouse conditions, the MIC (75 μg/mL) of silver nanoparticles was assessed to inhibit root rot diseases of beans by pre- and post-emergence damping off, root rot, and foliar disease severity. The study revealed that treating bean seeds with silver nanoparticles (AgNPs) at MIC of 75 μg/mL led to a decrease in damping off incidence and disease severity in pots infected with *M. phasolina, S. rolfsii, R. solani*, and *P. graminicola* compared to untreated bean seeds (Table 4). The fungi that cause root rot affect the growth and biomass of many plants, greatly affecting their production [43,63]. The results demonstrated that bean seeds treated with silver nanoparticles at MIC of 75 μg/mL exhibited enhanced germination rates, root length, shoot length, fresh weight, dry weight, and vigor index, compared to the control group (Figure 12). The germination rates, root length, shoot length, fresh weight, dry weight, and vigor index of bean seedlings were 100.00 cm, 55.00 cm, 35.33 cm, 61.33 g, 4.83 g, and 9033.33, respectively. In the case of control treatments, the results were 100, 00 cm, 45.00 cm, 26.00 cm, 50.00 g, 3.90 g, and 7100.00, respectively. In addition, the MIC of AgNPs played a major role in suppressing root rot pathogens, as seedlings that were treated with MIC and grown in pots inoculated with the pathogens showed much better growth rates than those that were not treated and grown in the same soil (Table 5). Consistent with our findings, the synthesized AgNPs significantly increased the root length, shoot length, fresh weight, and dry weight of rice seedlings [22]. On the other hand, while several studies, including our own, have shown positive effects of AgNPs on promoting plant growth, concerns have also been raised regarding their potential toxicity. It is speculated that AgNPs may exert a toxic effect on plants by affecting various aspects of plant physiology. These effects may include a reduction in the efficiency of chlorophyll absorption, a decrease in photosynthetic efficiency, a disturbance in nutrient absorption and transport, an alteration of hormone levels, a decrease in transpiration rates, and the potential to disrupt essential plant processes [64,65]. All of these concerns must be taken into consideration before expanding the use of nanomaterials in general and nano-silver in particular and moving into the field.

## 4. Conclusions

A safe alternative to traditional chemical synthesis techniques is the biosynthesis of silver nanoparticles (AgNPs) using plants. In this study, novel plant extracts were used to produce AgNP biosynthesis. Using XRD, FTIR, TEM, SEM, EDS, UV-vis spectroscopy, and other techniques, the creation of biogenic AgNPs was further verified and described. Moreover, in in vitro and greenhouse settings, biosynthesized AgNPs demonstrated potent antifungal action against root rot pathogens (*M. phaseolina, S. rolfsii, R. solani*, and *P. graminicola*) and enhanced the growth of bean plants. Therefore, these nanoparticles may play a vital role in protecting plants from infection by phytopathogenic fungi as a powerful alternative to fungicides and their harmful effects.

## Figures and Tables

**Figure 1 nanomaterials-14-00710-f001:**
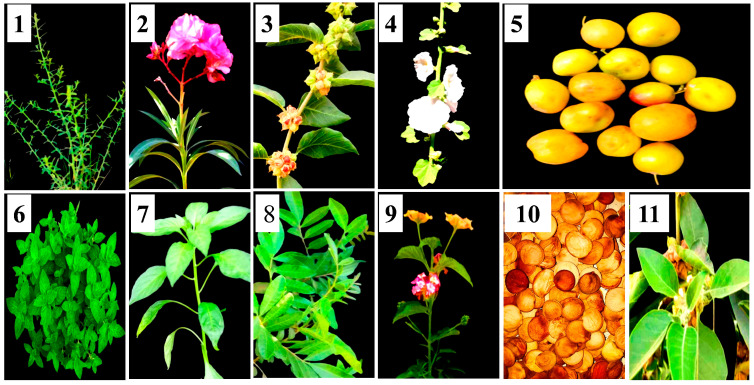
Plant parts used in the biosynthesis assay. (**1**) *Alhagi graecorum*, (**2**) *Nerium oleander*, (**3**) *Withania somnifera* (fruits), (**4**) *Althoea officinalis*, (**5**) *Ziziphus spina shristi*, (**6**) *Mentha arvensis*, (**7**) *Capsicum annuum*, (**8**) *Schinus terebinthifolius*, (**9**) *Lantana camara*, (**10**) *Bauhinia variegate*, and (**11**) *Withania smnifera* (leaves).

**Figure 2 nanomaterials-14-00710-f002:**
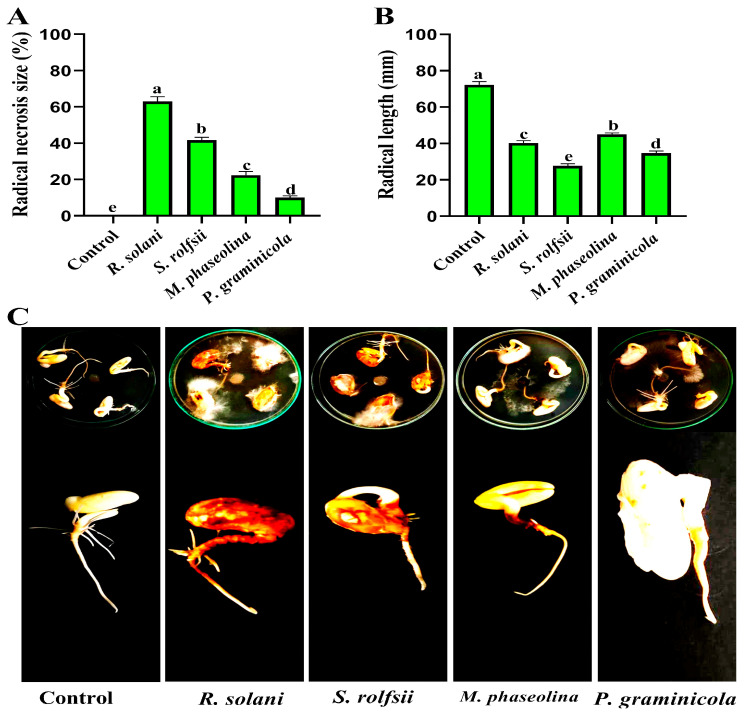
*R. solani, S. rolfsii, M. phaseolina*, and *P. graminicola* in vitro pathogenicity assay on been seeds. (**A**) The radical bean necrosis (%). (**B**) Effect of pathogenic root rot fungi on radical length. (**C**) Seeds infected with pathogens.

**Figure 3 nanomaterials-14-00710-f003:**
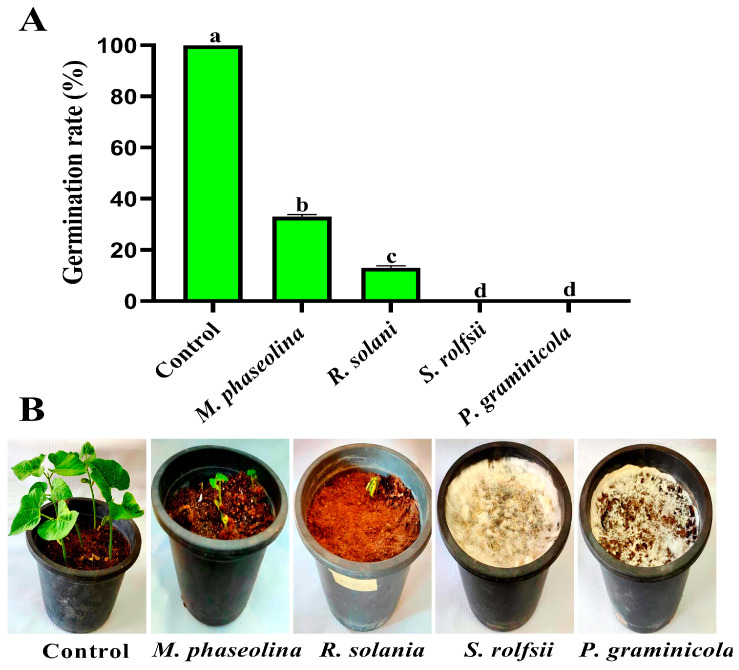
Pathogenicity assay of *M. phaseolina, R. solani, S. rolfsii*, and *P. graminicola* using the soil infection method under greenhouse conditions. (**A**) Germination rate (%) of bean seeds. (**B**) Pots treated and untreated by bean root rot fungi.

**Figure 4 nanomaterials-14-00710-f004:**
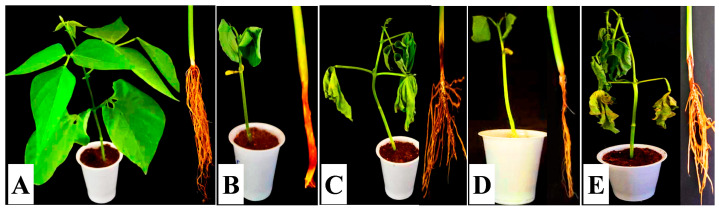
Pathogenicity assay using the seedling infection method under greenhouse conditions. (**A**) Control seedlings. (**B**) Seedlings inoculated with *S. rolfsii*. (**C**) Seedlings inoculated with *P. graminicola*. (**D**) Seedlings inoculated with *M. phaseolina*. (**E**) Seedlings inoculated with *R. solani*.

**Figure 5 nanomaterials-14-00710-f005:**
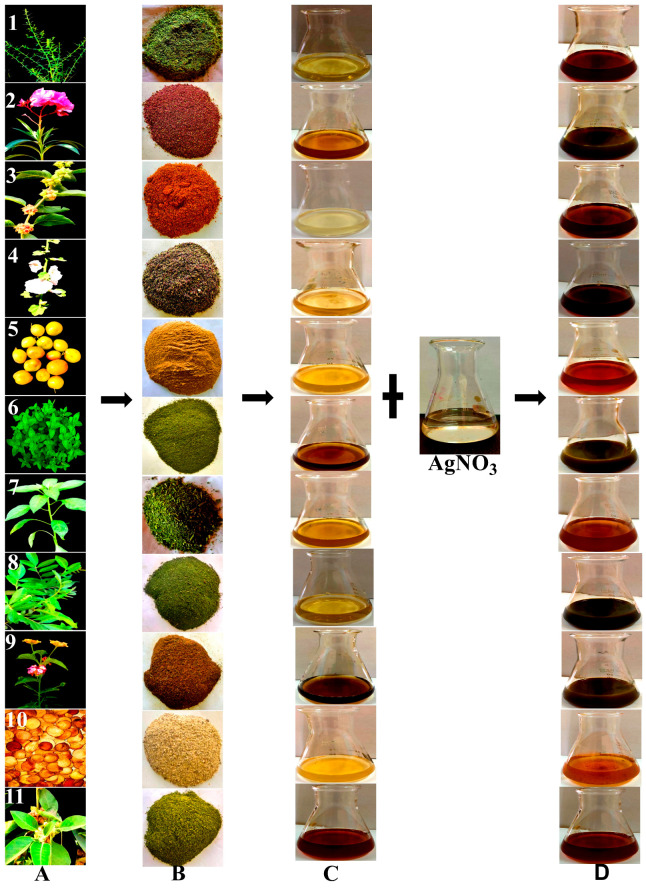
Confirmation of AgNP biosynthesis using plant extract. (**A**) The eleven plant parts were used in biosynthesis. (**A1**) *Alhagi graecorum*. (**A2**) *Nerium oleander*. (**A3**) *Withania somnifera* (fruits). (**A4**) *Althoea officinalis*. (**A5**) *Ziziphus spina shristi*. (**A6**) *Mentha arvensis*. (**A7**) *Capsicum annuum*. (**A8**) *Schinus terebinthifolius*. (**A9**) *Lantana camara*. (**A10**) *Bauhinia variegate*. (**A11**) *Withania smnifera* (leaves). (**B**) Plant powders. (**C**) Plant extracts. (**D**) The color changes from light yellow to dark brown following the incubation of 30 mL of each plant extract with 70 mL of AgNO_3_.

**Figure 6 nanomaterials-14-00710-f006:**
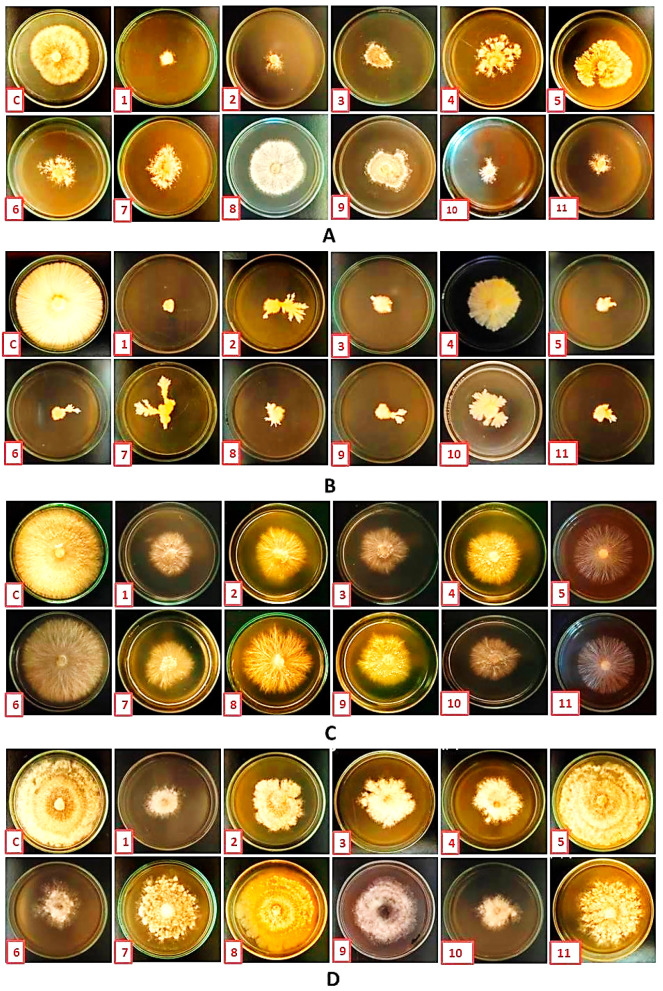
The antifungal activity of biosynthesized silver nanoparticles (AgNPs) at a concentration of 100 μg/mL evaluation against *R. solani* (**A**), *P. graminicola* (**B**), *S. rolfsii* (**C**), and *M. phaseolina* (**D**). The AgNPs were derived from various plant sources, namely *Alhagi graecorum* (**1**), *Nerium oleander* (**2**), *Withania somnifera* (fruits) (**3**), *Althoea officinalis* (**4**), *Ziziphus spina shristi* (**5**), *Mentha arvensis* (**6**), *Capsicum annuum* (**7**), *Schinus terebinthifolius* (**8**), Lantana *camara* (**9**), *Bauhinia variegate* (**10**), *Withania smnifera* (leaves) (**11**), and control (**c**).

**Figure 7 nanomaterials-14-00710-f007:**
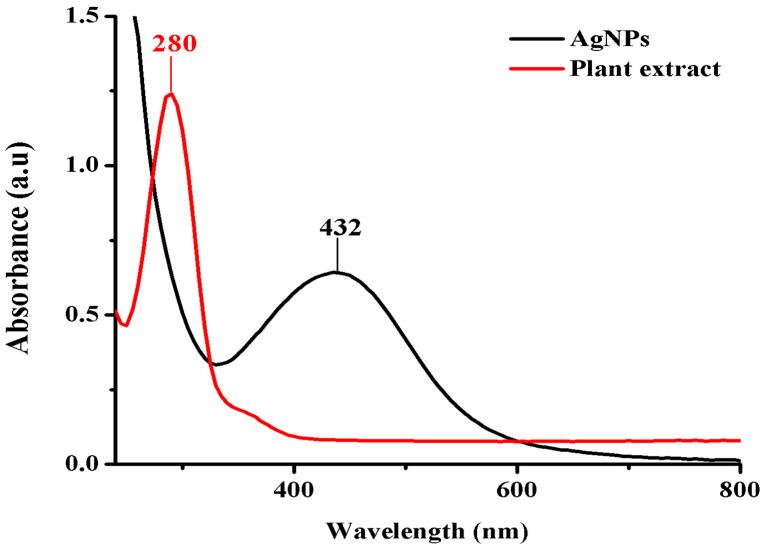
The UV-Vis spectrum of AgNPs synthesized using *A. graecorum* plant extract.

**Figure 8 nanomaterials-14-00710-f008:**
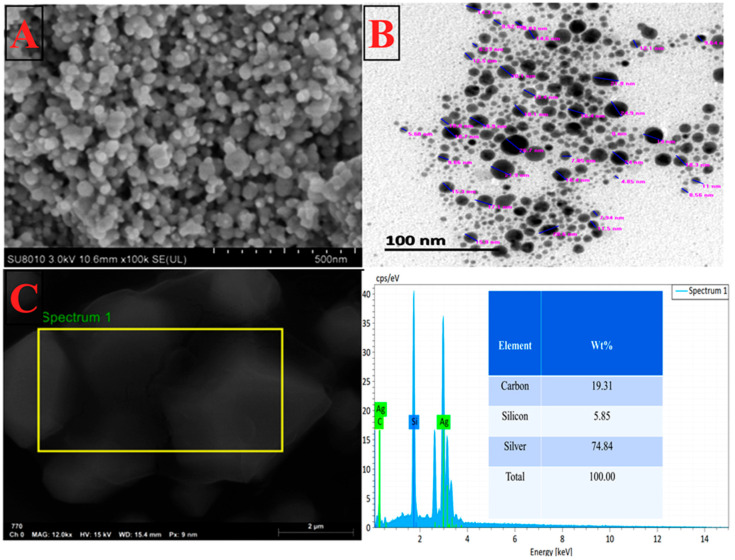
Characterization of AgNPs biosynthesized by using *A. graecorum* plant extract. (**A**) Scanning electron microscopy. (**B**) Transmission electron microscopy. (**C**) EDS analysis.

**Figure 9 nanomaterials-14-00710-f009:**
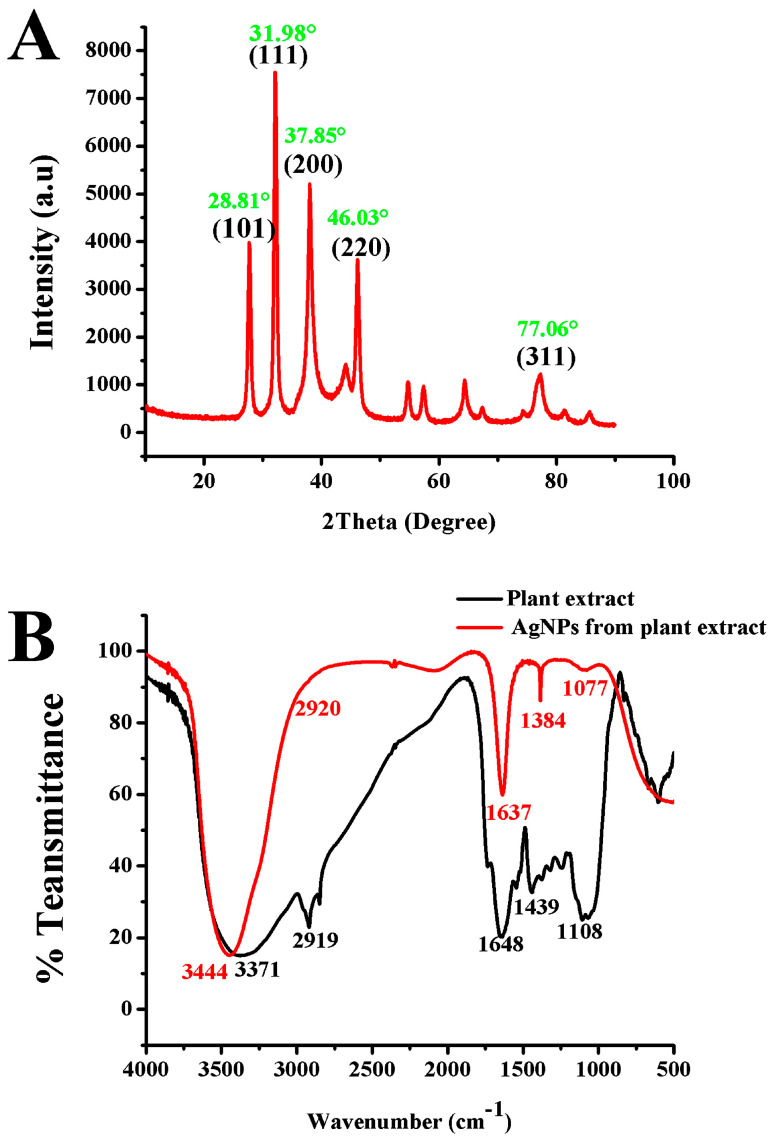
Characterization of the biosynthesized AgNPs using *A*. *graecorum* plant extract with X-ray diffraction (XRD) spectra (**A**) and Fourier-transform infrared (FTIR) spectra (**B**).

**Figure 10 nanomaterials-14-00710-f010:**
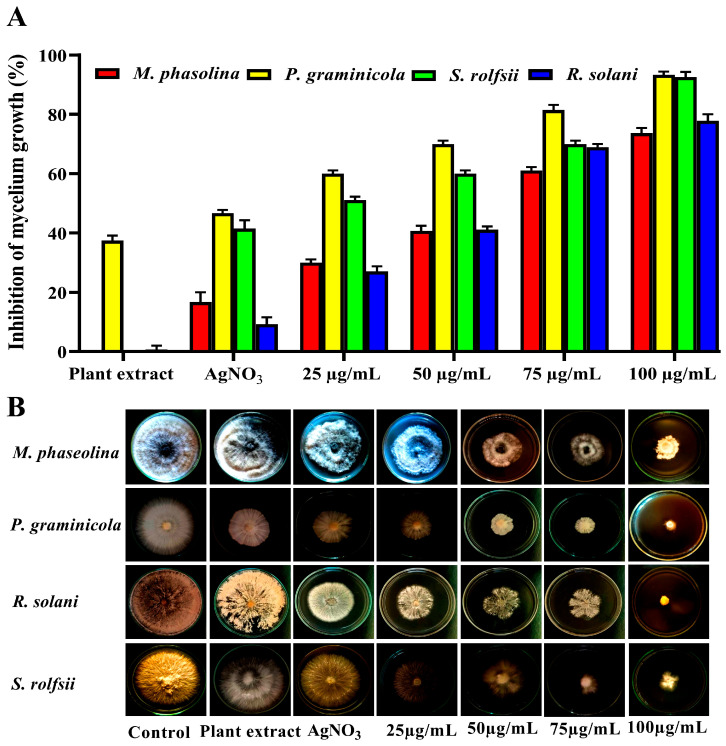
MIC assay of AgNPs against *M. phasolina, R. solani, S. rolfsii*, and *P. graminicola* causing bean root rot disease. Mycelium growth inhibition (%) (**A**). Radial development of mycelium on PDA medium (**B**).

**Figure 11 nanomaterials-14-00710-f011:**
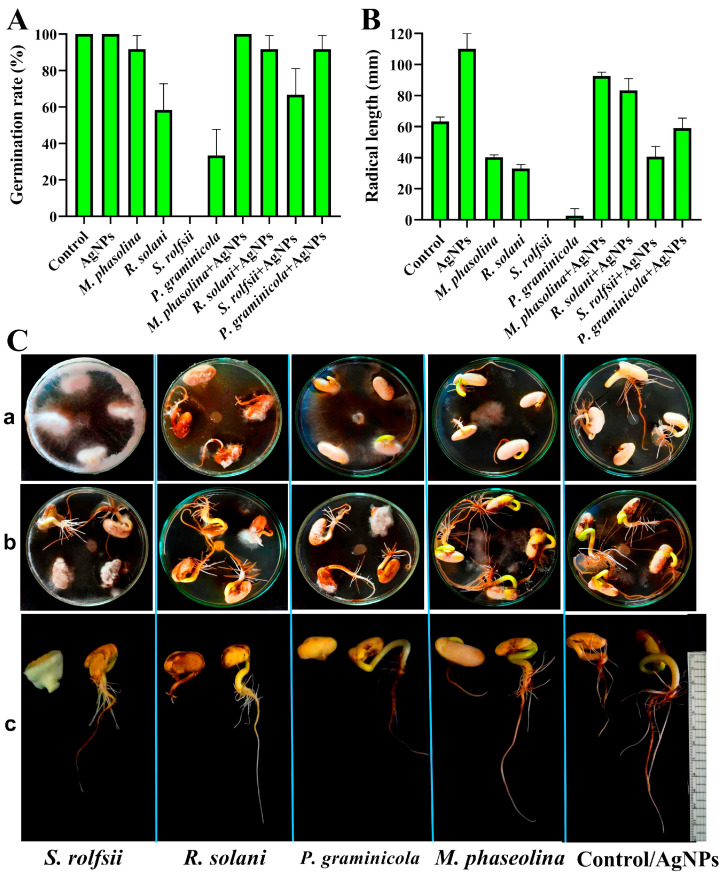
In vitro, the MIC (75 μg/mL) of AgNPs inhibited root rot diseases in beans by measuring the germination rates (%) (**A**) and radical length (**B**). Bean seeds grown in inoculated dishes with *M. phasolina, S. rolfsii, R. solani*, and *P. graminicola* (**Ca**), germination rates of bean seeds treated with AgNPs at a concentration of 75 μg/mL grown in inoculated dishes with pathogens (**Cb**), and the radical length of seeds treated (right) and untreated (left) with AgNPs grown in inoculated dishes with pathogens (**Cc**).

**Figure 12 nanomaterials-14-00710-f012:**
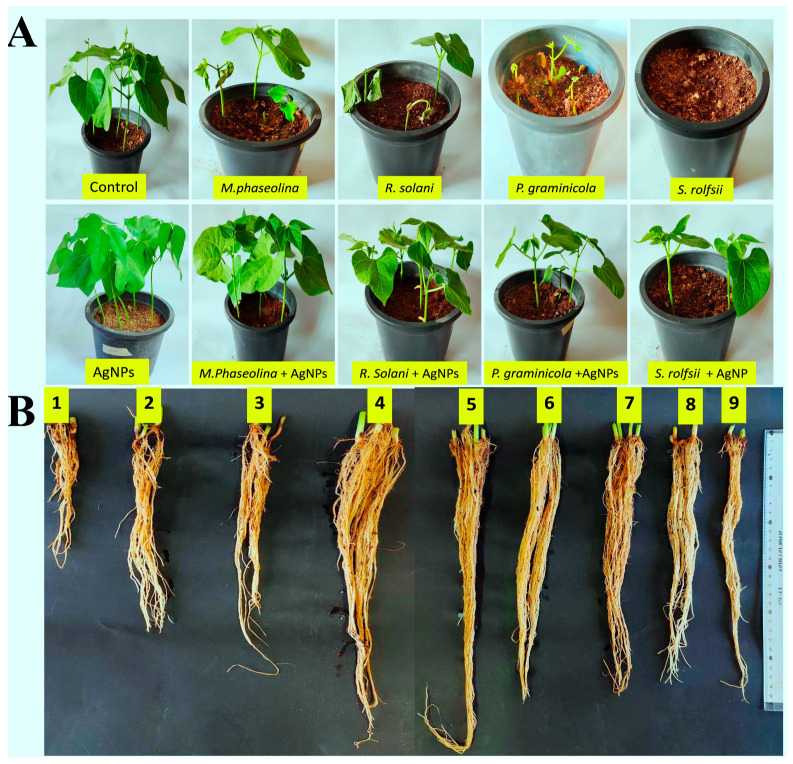
Root rot fungi inhibition using the MIC of silver nanoparticles under greenhouse conditions. (**A**) Seed beans treated and untreated with silver nanoparticles at the MIC of 75 μg/mL. (**B**) The root length of treated and untreated seed beans with *P. graminicola* (1), *R. solani* (2), *M. phasolina* (3), Control (4), AgNPs (5), *M. phasolina* +AgNPs (6), *R. solani* +AgNPs (7), *P. graminicola* +AgNPs (8), and *S. rolfsii* +AgNPs (9).

**Table 1 nanomaterials-14-00710-t001:** List of the eleven plants’ names in both English and Latin, along with the specific portions of each that are utilized in the biosynthesis of AgNPs.

English Name	Latin Name	Used Part
1- Manna tree	*Alhagi graecorum*	Leaves
2- Oleander	*Nerium oleander*	Flowers
3- Winter cherry	*Withania somnifera*	Fruits
4- Marshmallow	*Althoea officinalis*	Flowers
5- Christ’s thorn jujube	*Ziziphus spina shristi*	Leaves
6- Mint	*Mentha arvensis*	Leaves
7- Chili pepper.	*Capsicum annuum*	Fruits
8- Brazilian pepper-tree	*Schinus terebinthifolius*	Leaves
9- Lantanas	*Lantana camara*	Flowers
10- Orchid tree	*Bauhinia variegata*	Seeds
11- Winter cherry	*Withania smnifera*	Leaves

**Table 2 nanomaterials-14-00710-t002:** Pathogenicity assay of *M. phaseolina, R. solani, S. rolfsii*, and *P. graminicola* using the seedling infection method under greenhouse conditions.

Treatments	Disease Severity (%)		Growth Rates			
	Foliar Symptom	Root Rot Symptom	Shoot Length (cm)	Root Length (cm)	Fresh Weight (g)	Dry Weight (g)
*S. rolfsii*	100.00	100.00	11.00	0.00	0.40	0.20
*P. graminicola*	83.33	77.00	10.00	4.00	0.75	0.42
*R. solani*	55.00	38.00	26.20	7.00	1.44	0.61
*M. phaseolina*	44.14	32.00	28.00	8.00	2.10	0.80
Control	0.00	0.00	33.00	15.00	3.50	1.30

**Table 3 nanomaterials-14-00710-t003:** Antifungal activity of eleven silver nanoparticles synthesized using different plant extracts against bean root rot disease.

Treatments	Inhibition of Mycelium Growth (%)			
	*R. solani*	*P. graminicola*	*S. rolfsii*	*M. phaseolina*
1	92.60	94.44	75.93	79.63
2	89.61	85.56	62.96	63.33
3	62.52	90.00	70.74	66.67
4	51.21	31.48	61.11	60.00
5	15.70	93.33	55.19	3.33
6	60.08	91.85	8.89	68.52
7	55.65	76.67	68.89	35.19
8	14.19	87.78	26.30	2.22
9	53.68	90.00	56.67	30.00
10	87.24	53.70	72.22	75.19
11	83.71	92.22	52.22	37.41

(1) *Alhagi graecorum*; (2) *Nerium oleander*; (3) *Withania somnifera* (fruits); (4) *Althoea officinalis*; (5) *Ziziphus spina shristi*; (6) *Mentha arvensis*; (7) *Capsicum annuum*; (8) *Schinus terebinthifolius*; (9) *Lantana camara*; (10) *Bauhinia variegate*; (11) *Withania smnifera* (leaves).

**Table 4 nanomaterials-14-00710-t004:** Antifungal activities of AgNPs synthesized by *A. graecorum* against root rot fungi of beans under greenhouse.

Treatments	Disease Assessment			
	Pre (%)	Post (%)	FS (%)	RS (%)
*M. phasolina*	33.30	49.67	51.00	36.00
*R. solani*	66.70	83.33	59.33	43.67
*P. graminicola*	66.70	83.33	79.33	78.33
*S. rolfsii*	100.00	100.00	100.00	100.00
*M. phasolina* + AgNPs	0.00	8.33	12.67	6.67
*R. solani* + AgNPs	8.30	16.67	34.67	24.67
*P. graminicola* + AgNPs	16.70	19.33	54.00	43.00
*S. rolfsii* + AgNPs	58.30	50.00	70.00	63.00
AgNPs	0.00	0.00	0.00	0.00
Control	0.00	0.00	0.00	0.00

Pre = pre-emergence damping off (%), Post = post-emergence damping off (%), FS = foliar disease severity (%), and RS = root disease severity.

**Table 5 nanomaterials-14-00710-t005:** Effects of the biosynthesized AgNPs by *A. graecorum* on the growth parameters of beans.

Treatments	Growth Parameters					
	Sg (%)	Rl (cm)	Sl (cm)	Vi	Fw (g)	Dw (g)
*M. phasolina*	66.67	16.00	13.33	1955.56	14.53	1.03
*R. solani*	33.33	15.00	16.00	1033.33	13.53	0.70
*P. graminicola*	33.33	12.33	8.00	677.78	6.17	0.40
*S. rolfsii*	0.00	0.00	0.00	0.00	0.00	0.00
*M. phasolina* + AgNPs	100.00	28.33	20.33	4866.67	40.30	2.87
*R. solani* + AgNPs	91.67	31.33	24.00	5072.22	33.77	2.40
*P. graminicola* + AgNPs	83.33	28.67	22.00	4222.22	27.37	1.90
*S. rolfsii* + AgNPs	41.67	21.00	18.33	1638.89	20.83	1.27
AgNPs	100.00	55.00	35.33	9033.33	61.33	4.83
Control	100.00	45.00	26.00	7100.00	50.00	3.90

Sg = seed germination rates (%), Rl = root length (cm), Sl = shoot length (cm), Fw = fresh weight (g), DW = dry weight (g), and Vi = vigorous index.

## Data Availability

The data are contained within the manuscript.

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
