# Peer review of "Suppression of Root Rot Fungal Diseases in Common Beans (Phaseolus vulgaris L.) through the Application of Biologically Synthesized Silver Nanoparticles"

_nanomaterials, 2024, doi:10.3390/nano14080710_

Round 1
Reviewer 1 Report
Comments and Suggestions for Authors
The authors of "Suppression of Root Rot Fungal Diseases in Common Beans 2 (Phaseolus vulgaris L.) Through the Application of Biologically 3 Synthesized Silver Nanoparticles" showed an interesting insight into the food industry and its needs to improve production to be able to feed growing populations, the challenges ahead and possible solution with AgNP. The authors did huge efforts in using different plant extracts and synthesizing AgNP.
Following comments need to be addressed:
1-Line 78-79 not clear. There are plant extract mediated synthesis methods of AgNP, as the sentence before is anyhow acknowledging. There express better, what you exactly mean with "However, the possibility of using plant extracts as a biological agent 78 for the synthesis of nanoparticles and their use in reducing the above-mentioned diseases 79 has not yet been verified"
2-Figure 1 should be centered. Most of the pictures are not clear and of bad quality. Be careful of copyright issues.
3-AgNP have no long-term studies regarding their effects on soil, plants, aquatic life, ground water resources and human/animal health. There are already investigations about the toxicity of AgNP in the environment/soil/plants and their detrimental effects on the food chain. A critical viewpoint about AgNP use is completely missing. The authors need to include the disadvantages of AgNP into the manuscript and show possible solutions to the problem in synergy with their results. For now, the manuscript is just praising the advantages and effectivity of AgNP but completely lacks a critical evaluation. What about AgNP as a sustainable solution, what are the parameters needed to ensure safety for the coming generations ? How to use AgNP, if requred, in a more responsible and sustainable way? What are the economical and ecological implications of a widespread use of AgNP ? Where will be AgNP stored ? In leaves and seeds as well, which will lead to increase in AgNP levels in the human organisms after intake ? A critical parapgraph stating alll these concerns must be added.
4-How fast can the already existing resistance towards AgNP accelerated, if AgNP is used widespread on crops in huge amounts globally? Currently, some studies already point out an ongoing resistance of microorganisms against AgNP. Please refer to those studies in further paragraphs.
5-The references section shows irregularities and non-homogenous citing. Please ensure to use mdpi style in all references.
6-The introduction needs more about the critical outlook towards AgNP
Comments on the Quality of English Language
English is fine, needs only minor editing
Author Response
Response to Reviewer 1 Comments
The authors of "Suppression of Root Rot Fungal Diseases in Common Beans (Phaseolus vulgaris L.) Through the Application of Biologically Synthesized Silver Nanoparticles" showed an interesting insight into the food industry and its needs to improve production to be able to feed growing populations, the challenges ahead and possible solution with AgNP. The authors did huge efforts in using different plant extracts and synthesizing AgNP.
Response: We are thankful to the reviewer 1 for acknowledging our manuscript and suggesting the changes for further improvement. Indeed, this manuscript has been revised and improved based on the comments of the editor and the reviewers. In addition, the point-by-point response to each comment from Reviewer 1 is listed below.
Point 1: Line 78-79 not clear. There are plant extract mediated synthesis methods of AgNP, as the sentence before is anyhow acknowledging. There express better, what you exactly mean with "However, the possibility of using plant extracts as a biological agent 78 for the synthesis of nanoparticles and their use in reducing the above-mentioned diseases 79 has not yet been verified"
Response: Thanks for the valuable suggestion. The text has been rewritten correctly (highlighted in yellow; see Lines No. 108 to 110).
Point 2: Figure 1 should be centered. Most of the pictures are not clear and of bad quality.
Response: Thank you for your valuable suggestion. We greatly appreciate it. We have made significant improvements to the quality of Figure 1, as well as enhanced most of the Figures included in the manuscript.
Point 3: AgNP have no long-term studies regarding their effects on soil, plants, aquatic life, ground water resources and human/animal health. There are already investigations about the toxicity of AgNP in the environment/soil/plants and their detrimental effects on the food chain. A critical viewpoint about AgNP use is completely missing. The authors need to include the disadvantages of AgNP into the manuscript and show possible solutions to the problem in synergy with their results. For now, the manuscript is just praising the advantages and effectivity of AgNP but completely lacks a critical evaluation. What about AgNP as a sustainable solution, what are the parameters needed to ensure safety for the coming generations ? How to use AgNP, if requred, in a more responsible and sustainable way? What are the economical and ecological implications of a widespread use of AgNP ? Where will be AgNP stored ? In leaves and seeds as well, which will lead to increase in AgNP levels in the human organisms after intake ? A critical parapgraph stating alll these concerns must be added.
Response: Many thanks for your valuable comments. Concerns rose about applications of AgNPs in the agricultural field and their toxic effects are included. (Highlighted in yellow, see Line 83-92 and line 422 - 451).
Point 4: How fast can the already existing resistance towards AgNP accelerated, if AgNP is used widespread on crops in huge amounts globally? Currently, some studies already point out an ongoing resistance of microorganisms against AgNP. Please refer to those studies in further paragraphs.
Response: Thank you very much for your valuable suggestion. Some studies have been mentioned that show the development of resistance to some microbes toward silver nanoparticles in the text (Highlighted in yellow, see Line 409-415).
Point 5: The references section shows irregularities and non-homogenous citing. Please ensure to use mdpi style in all references.
Response: Thank you very much for your comment. All references have been carefully checked in keeping with the format and style of the journal.
Point 6: The introduction needs more about the critical outlook towards AgNP
Response: Many thanks for your valuable comments.. We have now included the (Highlighted in yellow, see Line 74-110).

Reviewer 2 Report
Comments and Suggestions for Authors
The use of plant extracts for synthesising nanoparticles is becoming more popular due to its environmentally friendly and sustainable approach. Silver nanoparticles (AgNPs) are of particular interest due to their unique properties, including antimicrobial activity, catalytic behaviour, and optical properties. Plant extracts, which are rich in bioactive compounds, act as reducing and stabilising agents during the synthesis of AgNPs. In the submitted paper titled 'Suppression of Root Rot Fungal Diseases in Common Beans (Phaseolus vulgaris L.) Through the Application of Biologically Synthesized Silver Nanoparticles', the authors evaluated the antifungal activity of biosynthesized AgNPs against fungi causing root rot in common beans both in vitro and under greenhouse conditions.
The authors' work is impressive both quantitatively and qualitatively. However, the text has some shortcomings that I hope the authors will correct quickly.
In Chapter 2 (Materials and Methods), all reagents used should be listed.
In Chapter 2.3.2, it is unclear whether the authors controlled the pH or simply mixed the resulting extracts with AgNO3 without modifying the environment.
The authors only provide UV-Vis spectra for AgNPs synthesized with A. graecorum, but it would be beneficial to compare with others. The same applies to electron microscopy. Additionally, due to the colour of the extracts, it would be relevant to conduct UV-Vis spectra for the extracts. What is the cause of the spectrum jerking between 490-540 nm?
Although it may be challenging to post numerous microscope images, some form of comparison would be highly beneficial.
Sentence (line 315-318) ‘The peak at 565 cm-1 is due to the C-Br elongation, which is characteristic of alkyl halides. The presence of such groups in the chlorofluorocarbons (CFCs) from the plant extract of A. graecorum confirms the presence of proteins and indicates that these functional groups play a major role in reducing Ag+ to Ag0’ is quite general AND debatable. It is difficult to talk about a peak at 565 cm-1.
However, it is unclear which specific functional groups are responsible for this reduction. To draw comparable conclusions, it is necessary to compare the FTIR spectra of the extracts before and after the reduction of silver ions.
The functional tests are acceptable, but the discussion of the physicochemical properties requires improvement. It would be valuable to compare and discuss the results obtained using different methods, such as SEM with XRD or UV-Vis with SEM. The conclusions need to be rephrased and the key findings should be emphasised.
Author Response
Response to Reviewer 2 Comments
The use of plant extracts for synthesising nanoparticles is becoming more popular due to its environmentally friendly and sustainable approach. Silver nanoparticles (AgNPs) are of particular interest due to their unique properties, including antimicrobial activity, catalytic behaviour, and optical properties. Plant extracts, which are rich in bioactive compounds, act as reducing and stabilising agents during the synthesis of AgNPs. In the submitted paper titled 'Suppression of Root Rot Fungal Diseases in Common Beans (Phaseolus vulgaris L.) Through the Application of Biologically Synthesized Silver Nanoparticles', the authors evaluated the antifungal activity of biosynthesized AgNPs against fungi causing root rot in common beans both in vitro and under greenhouse conditions.
The authors' work is impressive both quantitatively and qualitatively. However, the text has some shortcomings that I hope the authors will correct quickly.
Response: We are thankful to the reviewer 2 for acknowledging our manuscript and suggesting the changes for further improvement of the manuscript. The point-by-point response to each comment is listed below.
Point 1: In Chapter 2 (Materials and Methods), all reagents used should be listed.
Response: Thank you very much for your valuable suggestion. In this study, all reagents used have been listed. Also, the components of the media used in the experiments have been indicated, along with the respective sources of these ingredients. (Highlighted in yellow, see Line 116-117 and 169-170).
Point 2: In Chapter 2.3.2, it is unclear whether the authors controlled the pH or simply mixed the resulting extracts with AgNO3 without modifying the environment?
Response: I am grateful for your insightful comments. Indeed, the pH of the mixture (silver nitrate particles and plant extract) was set to 7, and this was indicated in the text. Several studies indicate that PH7 d is considered the most suitable for silver nitrate reduction (Rashidipur et al., 2014). (Highlighted in yellow, see Line 171).
Point 3: The authors only provide UV-Vis spectra for AgNPs synthesized with A. graecorum, but it would be beneficial to compare with others. The same applies to electron microscopy. Additionally, due to the colour of the extracts, it would be relevant to conduct UV-Vis spectra for the extracts. What is the cause of the spectrum jerking between 490-540 nm?
Response: Thank you for your valuable suggestion. In this study, we conducted the synthesis of eleven different silver nanoparticle compounds using various plant extracts. These compounds were evaluated for their efficacy in suppressing root-rot pathogens in beans. The findings revealed that AgNPs synthesized by A. graecorum exhibited the highest effectiveness in inhibiting and suppressing root rot pathogens. Consequently, this particular nanocomposite was selected for further investigation, including the characterization of its properties using UV, TEM, SEM, XRD, and FTIR techniques. In addition to conducting more experiments that clarify its role in inhibiting root rot pathogens.
To streamline the research process and optimize resources, we initially explored the potential of these nanomaterials for inhibiting plant pathogens. The best one of them was subjected to comprehensive studies, including characterization, to expedite progress while minimizing time, effort, and cost.
Moreover, in response to your suggestion, we have included a comparison between the UV spectra of the synthesized silver nanoparticles using A. graecorum and the extract used for their synthesis. Please refer to Figure 7 for the relevant data.
Point 4: Although it may be challenging to post numerous microscope images, some form of comparison would be highly beneficial.
Response: Thank you for your valuable suggestion. As you correctly pointed out, capturing numerous SEM and TEM microscopic images of nanomaterials can be challenging. In light of this, we initially conducted a preliminary investigation involving eleven nanocomposites to assess their efficacy against root rot pathogens in the laboratory. Among these nanocomposites, the compound that exhibited the most potent inhibition was selected for further extensive experimentation, including the characterization of its properties. This approach was adopted to optimize time and effort, considering the high costs associated with conducting such experiments.
Point 5: Sentence (line 315-318) ‘The peak at 565 cm-1 is due to the C-Br elongation, which is characteristic of alkyl halides. The presence of such groups in the chlorofluorocarbons (CFCs) from the plant extract of A. graecorum confirms the presence of proteins and indicates that these functional groups play a major role in reducing Ag+ to Ag0’ is quite general AND debatable. It is difficult to talk about a peak at 565 cm-1.
Response: Thank you very much for your great suggestion. This paragraph has been rewritten again (Highlighted in yellow, see Line 349- 369).
Point 6: However, it is unclear which specific functional groups are responsible for this reduction. To draw comparable conclusions, it is necessary to compare the FTIR spectra of the extracts before and after the reduction of silver ions.
Response: Thank you very much for your great suggestion. In Figure 9B, we compare the FTIR spectra of the extracts before and after silver ion reduction according to your suggestion.
Point 7 The functional tests are acceptable, but the discussion of the physicochemical properties requires improvement. It would be valuable to compare and discuss the results obtained using different methods, such as SEM with XRD or UV-Vis with SEM. The conclusions need to be rephrased and the key findings should be emphasised.
Response: Thank you very much for your great suggestion. The physicochemical properties are discussed, and the results obtained from UV, SEM, TEM, XRD, and FTIR are presented better than the previous image. (Highlighted in yellow, see Line 310- 369).

Round 2
Reviewer 1 Report
Comments and Suggestions for Authors
The authors added the needed comments.
Sentence 106-107
Therefore, this study aims to use plant extracts to synthesize AgNPs and evaluate their anti-AgNP effectiveness for fungi against fungal diseases of root rot in beans.
Is in the second part still unclear.
It must be "Therefore, this study aims to use plant extracts to synthesize AgNPs and evaluate their anti-fungal activities against diseases of root rot in beans. "
line 307: Synthsized with must be "synthesized"
Comments on the Quality of English LanguageAll fine, except that sentence line 106-107.
Can be corrected and prepared for publication
Author Response
The authors added the needed comments.
Response: We sincerely appreciate Reviewer 1 for dedicating time to review our manuscript and for providing valuable comments that have significantly enhanced the quality of the paper. We are grateful for his valuable input, which has contributed to the overall improvement of the manuscript.
Point 1: Sentence 106-107
Therefore, this study aims to use plant extracts to synthesize AgNPs and evaluate their anti-AgNP effectiveness for fungi against fungal diseases of root rot in beans.
Is in the second part still unclear.
It must be "Therefore, this study aims to use plant extracts to synthesize AgNPs and evaluate their anti-fungal activities against diseases of root rot in beans.”
Response: The sentence was written exactly as you suggested. (Highlighted in yellow; see Lines No. 108 to 110).
Point 2: line 307: Synthsized with must be "synthesized".
Response: Sorry for this mistake. Synthsized has been replaced by Synthesized. (Highlighted in yellow; see Lines No. 309).

Reviewer 2 Report
Comments and Suggestions for Authors
The authors have responded to the questions and suggestions. I believe that the article is suitable for publication in its present form.
Author Response
The authors have responded to the questions and suggestions. I believe that the article is suitable for publication in its present form.
Response: We sincerely appreciate Reviewer 2 for dedicating time to review our manuscript and for providing valuable comments that have significantly enhanced the quality of the paper. We are grateful for his valuable input, which has contributed to the overall improvement of the manuscript.
